# Effect of Mineral Nutrition and Salt Spray on Cucumber Downy Mildew (*Pseudoperonospora cubensis*)

**DOI:** 10.3390/plants11081007

**Published:** 2022-04-07

**Authors:** Dor Barnea, Uri Yermiyahu, Dalia Rav-David, Yigal Elad

**Affiliations:** 1Department Plant Pathology and Weed Research, Agricultural Research Organization, The Volcani Institute, 68 Hamakabim Rd, Rishon LeZion 7534509, Israel; dor.barnea@gmail.com (D.B.); dalia@volcani.agri.gov.il (D.R.-D.); 2The Robert H. Smith Faculty of Agriculture, Food and Environment, The Hebrew University of Jerusalem, Rehovot 7610001, Israel; 3Agricultural Research Organization, Gilat Research Center, D.N. Negev 2, Bet-Dagan 8528001, Israel; uri4@volcani.agri.gov.il

**Keywords:** agrotechnical control, chloride, *Cucumis sativus*, cultural control, downy mildew, integrated management, magnesium, plant disease, potassium

## Abstract

It was previously shown that spraying with CaCl_2_, MgCl_2_, KCl, and K_2_SO_4_ and high N and Mg concentrations in the irrigation water of potted cucumber plants reduced powdery mildew, while medium P and high K concentrations increased powdery mildew. In the present work, we tested the effect of irrigation with N, P, K, Ca, and Mg and spraying with salts on downy mildew (*Pseudoperonospora cubensis*) of cucumber (CDM). In potted plants, an increase in N concentration in the irrigation water resulted in a major increase in CDM severity, while an increase in K or Ca concentrations resulted in a gradual increase in CDM severity. An increase in P and Mg concentrations in the irrigation water resulted in a major CDM decrease. Spraying with Ca, Mg, and K salts with Cl and SO_4_ anions resulted in CDM suppression in most cases, and a negative correlation was obtained between the salt and anion molar concentrations and the CDM severity. Using NaCl sprays, both Na and Cl concentrations were negatively related to the CDM severity. MgCl_2_ (0.1 M Cl), K_2_SO_4_ (0.1 M SO_4_), MgCl_2_ + K_2_SO_4_, and monopotassium phosphate (MKP, 1%) sprayed under commercial-like (CL) conditions significantly reduced CDM by 36.6% to 62.6% in one disease cycle, while, in a second cycle, CDM was significantly reduced only by K_2_SO_4_ and MKP. In conclusion, fertigation with P and Mg, and salt spraying decreased CDM, while only spraying under CL resulted in CDM suppression.

## 1. Introduction

Nutritional elements are essential for plant growth [1,2] and, for a long time, they have been recognized for their effect on plant health [3,4,5]. The nutrient status of plant tissues essentially influences plant growth and productivity, pathogen infection, colonization, and sporulation [6]. Furthermore, the nutritional elements may increase or decrease the susceptibility of plants to certain diseases when used at various concentrations [5,7,8]. Plant nutrition may be administrated as part of the fertilization regime with irrigation water or as water-dissolved salts that are applied by spraying [1]; both approaches may affect plant diseases as demonstrated in recent studies with sweet basil diseases [8,9] and with cucumber powdery mildew [7]. The studies on sweet basil diseases caused by *Sclerotinia sclerotiorum* [10], *Botrytis cinerea* [11], *Peronospora belbahrii* [8,12], and tomato [13] revealed different effects of N, P, K, Ca, and Mg, depending on the plant host and the pathogen involved.

The present work deals with downy mildew of cucumber (CDM) caused by the oomycete *Pseuperonospora cubensis*. It is a disease promoted by high humidity that causes severe leaf damage, and the pathogen requires a film of water to establish infection. It is commonly managed by cultural means that minimize the presence of water on the leaves, host resistance, and the intensive use of chemical pesticides [14,15,16,17]. 

Nutritional elements can potentially affect downy mildew (DM). The severity of sweet basil DM (*P. belbahrii*) increased with high total N concentration in the irrigation water and the shoots, as well as when a high rate of 30–40% of NH_4_^+^ for the total N in the irrigation water was supplied [12]. An increased K concentration in the irrigation water of sweet basil increased DM severity, and sprays of KCl and K_2_SO_4_ reduced the disease. High Ca or Mg in the irrigation water and CaCl_2_ spray decreased the DM severity [8]. Interestingly, microelements affected sweet basil DM as well; sprayed Mn-EDTA and Zn-EDTA decreased the DM severity of sweet basil, and their application in the irrigation water reduced DM by more than 50% [9].

There is limited information on the effect of nutrition on CDM. N supplied as ammonium nitrate was applied at concentrations of 100–600 mg·L^−1^ in the irrigation water to potted cucumber plants. DM severity was 24% lower in the 300 mg·L^−1^ N treatment [18]. More information is available on the nutrition effect on cucurbit powdery mildew (*Podosphaera*
*xanthii*). For instance, cucumber powdery mildew was decreased by CaCl_2_ and MgCl_2_, or KCl and K_2_SO_4_ sprays. Furthermore, high N and Mg concentrations in the irrigation water reduced the disease in young cucumber plants, and a medium amount of P and a high amount of K increased its severity. Mature cucumber plants had less severe powdery mildew at higher N, lower P, and higher K levels [7]. Spray application of P and K salts controlled cucumber powdery mildew [19].

In the present study, we examined the effects of N, P, K, Ca, and Mg, applied in the irrigation water, and the effect of sprays with salts containing the cations Ca, Mg, and K and the anions Cl and SO_4_ on CDM. We initially tested in potted plants the effects of irrigation with the nutritional elements and spray treatments; later, we examined the effects of several salts and a salt combination on CDM under commercial-like conditions in a net house.

## 2. Results

### 2.1. Pot Experiments—Relationship between the Concentrations of Supplemental Nutrients in the Fertigation Solution and CDM Severity

#### 2.1.1. Relationship between N, P, and K Nutrition through the Fertigation Solution with CDM Severity

N, P, and K concentrations in the irrigation water were achieved by changing their concentrations without changing the concentrations of the other major ions, as described in Section 4. 

N (Expts B-N-f): Increasing the concentration of N in the fertigation solution (15% NH_4_^+^, 85% NO_3_^−^) from 0.7 to 14.3 mM resulted in a gradual increase in N concentration in the cucumber leaves up to 3.91% of the leaf dry weight (Figure 1a). An increase in the N concentration in the irrigation water resulted in a major increase in CDM severity at 5.0 and 12.4 mM N in the water (Figure 1b). Higher N concentrations in the leaves were associated with high CDM severity. Leaf N concentrations of 3.27–3.43% resulted in the highest severity, while lower severe disease severity was observed at the highest N concentration (3.72% of dry leaf weight) (Figure 1c). 

P (Expts B-P-f): Increasing the concentration of P in the fertigation solution from 0.065 to 0.65 mM resulted in concentrations of 0.21% and 0.79% P in the cucumber leaves. The change in P concentration in the irrigation water resulted in a major decrease in CDM severity (Figure 2a). Consequently, a decrease in CDM resulted from increased P concentration in the leaves (Figure 2b). 

K (Expts B-K-f): Increasing the concentration of K in the fertigation solution from 0.5 to 5.1 mM resulted in a gradual increase in K concentration in the cucumber leaves up to 4.79% of the leaf dry weight (Figure 3a). An increase in the K concentration in the irrigation water resulted in a gradual increase in CDM severity (Figure 3b). Similarly, the change in K leaf concentration resulted in a gradual increase in disease severity (Figure 3c). Calculation of the relationship between K concentration of individual plants and CDM severity resulted in a positive correlation (Results not presented).

#### 2.1.2. Relationship between Ca and Mg Nutrition through the Fertigation Solution with CDM Severity

The effect of Ca and Mg on CDM severity was tested by adding their Cl^−^ salts to the irrigation water without changing the concentrations of the other major ions (see Section 4). 

*Ca* (*Expts B-Ca-f*): Increasing the concentration of Ca in the fertigation solution from 1.0 to 4.0 mM resulted in a gradual increase in Ca concentration in the cucumber leaves up to 4.17% of the leaf dry weight (Figure 4a). An increase in the Ca concentration in the irrigation water increased CDM severity (Figure 4b). Similarly, the change in Ca leaf concentration increased disease severity, whereby the two higher concentrations of Ca were associated with a significantly higher CDM severity than the two lower concentrations (Figure 4c).

Mg (Expts B-Mg-f): Increasing the concentration of Mg in the fertigation solution from 0 to 4.94 mM resulted in a gradual increase in Mg concentration in the cucumber leaves up to 2.35% of the leaf dry weight (Figure 5a). An increase in the Mg concentration in the irrigation water resulted in a gradual decrease in CDM severity (Figure 5b). Similarly, the change in Mg leaf concentration resulted in a gradual decrease in disease severity (Figure 5c).

### 2.2. Spray of Salts for the Control of CDM

#### 2.2.1. Spray Application of Salts for CDM Suppression (Expt. A-s1)

Spraying of Ca, Mg, and K salts with Cl and SO_4_ in 0.5% to 1.0% concentrations was performed on mature cucumber plants. CDM developed on the leaves and was significantly suppressed by most salts (Figure 6a). The relationship between the molar concentration of the applied salts according to the cations and the anions of the salts and the CDM severity was calculated. A significant negative correlation was obtained between the anion-related molar concentrations of the salts and the severity of the disease. It can be seen that anion concentrations above 0.03 M resulted in significant CDM suppression (Figure 6b). Considering the effect of the cations, with one outlier disregarded, all other cation molar concentrations were significantly different from the 0 (control) concentration, and most anion concentrations were different from the control (results not presented).

#### 2.2.2. Effect of NaCl Spray on CDM (Expt. B-s1)

To evaluate the effect of the Cl concentrations of sprayed salts, we tested NaCl as the sole salt in spray over cucumber plants at concentrations of 0 to 0.6 M (Cl-wise). With the exception of one data point, 0.2–0.6 M NaCl significantly suppressed CDM by 74.8% and more (Figure 7a), as well as the entire epidemic development; according to AUDPC values, the spray with NaCl at 0.4–0.6 M suppressed CDM by 63–88%, respectively (Figure 7b). It can be considered that spraying with a higher concentration of Cl^−^ salt results in the suppression of CDM. 

#### 2.2.3. Effect of Na and Cl Concentrations in the Leaves on CDM Severity (Expt. B-s2)

NaCl spray on cucumber plants at concentrations of 0 to 0.6 M resulted in 0.30–1.22% Na and 0.35–2.25% Cl in the leaves (Figure 8a). Relations were drawn between each of the ion concentrations and the level of CDM severity. Both ions exhibited a negative relationship with the CDM severity (Figure 8b,c) essentially because of a significant correlation between the concentrations of the ions in the spray solution and in the treated leaves.

#### 2.2.4. Effect of the Spray of Cl and SO_4_ Salt Solutions on CDM of Cucumber Plants Grown in Pots (Expts B-s3)

Mg and K salts containing either Cl or SO_4_ anions were sprayed at a concentration of 0.2 M (anion-wise) (Figure 9). The salts effectively suppressed the CDM severity except for MgSO_4_. 

### 2.3. Semi-Commercial Experiments to Test the Effects of Spray Applications of Various Salts on CDM (Expts B-SCs-a/b)

The semi-commercial experiments involved mature, fruit-bearing plants and included (a) MgCl_2_, CaCl_2_, and K_2_SO_4_ at 0.05 and 0.1 M (Cl or SO_4_-related) and a combination of 0.05 M of these salts and 0.1 M K_2_(SO_4_) (Expt. B-SCs-b), or (b) sprays of MKP and polyhalite (Expt. B-SCs-a), which are commercially available as complex fertilizers and contain K or Mg, Ca, and S. The salts generally reduced CDM severity, and there was no better disease suppression upon combining the salt with applications of the same concentrations of the individual salts (Figure 10a). CDM severity was significantly reduced by MKP and polyhalite (Figure 10b).

### 2.4. Effect of Spray Treatments on CDM under Commercial-Like Conditions

Two CL experiments were carried out to test the effect of salts sprays on CDM. In the first experiment (CL1). There were two distinct disease cycles during the season of cucumber growth in the net house, where CDM symptoms appeared first on the lower leaves (first cycle) and later the on upper leaves (second cycle). MgCl_2_ + K_2_SO_4_ 0.1 M (Cl) + 0.1 M (SO_4_) significantly reduced CDM by 42.5% and 36.9% in the two disease cycles, respectively (Figure 11a,b). Cumulative fresh yield in the water control and the MgCl_2_ + K_2_SO_4_ treatment was 12.025, and 14.057 kg/10 plants, respectively, and the number of fruits in the respective treatments was 116.3 and 133.4 fruits/10 plants. Differences in both yield parameters were significant (*p* < 0.05).

In the second CL experiment (CL2), disease was significantly reduced during the first CDM cycle by all spray treatments (MgCl_2_ (0.1 M Cl), K_2_SO_4_ (0.1 M SO_4_^−^), MgCl_2_ + K_2_SO_4_ (as above), and MKP (1%)), achieving the values of disease reduction of 36.6%, 62.6%, 52.8%, and 55.1%, respectively (Figure 12a). In the second cycle of the same experiment (CL2), the disease was significantly reduced only by K_2_SO_4_ and MKP, achieving the values of disease reduction of 50.2% and 45.5%, respectively (Figure 12b). The weight of fresh cucumber fruits in the various treatments ranged between 12.416 and 15.994 kg/10 plants, and the number of fruits ranged between 120.8 and 143.3 fruits/10 plants. Nevertheless, the differences between treatments were insignificant; hence, the detailed yields are not presented.

## 3. Discussion

### 3.1. Application of Nutritional Elements in the Irrigation Water

Recently, we reported the effect of the nutritional elements N, P, K, Ca, and Mg on cucumber powdery mildew (CPM) [7]. Here, we report the effect on CDM by these nutritional elements that were applied via the irrigation water to the root zone or by spraying salts dissolved in water. The inclusion of nutritional elements in the irrigation water to affect the CDM revealed a different and unique outcome for each of the tested elements. An increase in N concentrations promoted the disease, although CDM was at a medium severity at the higher concentration. P at a higher concentration reduced CDM, while an increase in K and Ca concentrations increased the disease severity. Increased Mg concentrations were associated with decreased CDM severity in the cucumber leaves. N, P, and K concentration manipulation under commercial conditions did not result in a conclusive effect of any of the three elements (results not presented). 

It is interesting to compare the outcome of the irrigation with higher nutritional elements in potted cucumber plants between CDM and powdery mildew (PM). Unlike the N promoting effect on CDM, N partially decreased PM. Unlike the P suppressive effect and Ca promoting effect on CDM, P did not decrease PM, and Ca did not affect PM. High K concentrations enhanced both diseases on cucumber leaves, while Mg suppressed both diseases similarly [7]. In general, although similar treatments were administrated to cucumber plants, the general outcome, excluding Mg, was different for downy mildew (DM)and PM. Thus, it may be impossible to draw common conclusions about a common nutritional element that can be implemented in the irrigation water to suppress both diseases in cucumber.

The effect of nutritional elements on cucurbits DM and PM of additional crops varies with the pathosystem and even with the study. For instance, muskmelons grown in sand and irrigated with different nutrient solutions had lower DM on plants grown with high N (unlike our results with CDM), high P, and low K (similar to our results with CDM) [20]. Similar to the current CDM results, in the pathosystem of DM (*Peronospora alta*) on blond psyllium (*Plantago ovata*), omission of Mg led to increased disease intensity [21]. Increased N concentrations increased the severity of canola DM severity [22], and camelina (*Camelina sativa*) DM disease also increased with applied N rates [23]. On the contrary, unlike our high N promotion of CDM, nitrogen in the form of NH_4_NO_3_ fertilizer supplied to potted cucumber plants through irrigation water was described as a means of control (24% reduced severity) of the disease in cucumber [18]. The addition of Si and Ca reduced the CDM percentage of infected leaves and disease severity, a result that was not achieved for Ca in the current research [24]. Tomesh and Struckmeyer [25] reported that lower levels of N and Ca in cucumber plants were associated with more minor symptoms and fewer conidiophores of the PM pathogen.

### 3.2. Spraying of Salts over the Plant’s Canopy

Interestingly, sprays of salts on potted plants resulted in CDM suppression when applied both at low and high concentrations, with no major difference between cations (K^+^, Ca^2+^, Mg^2+^) at the two concentrations and anions (Cl^−^ vs. SO_4_^−2^) at 0.03 M concentration and above. Sprayed Na (Cl salt) also affected CDM but at a higher molar concentration than the other salts; therefore, it was generally less effective as compared with these other ions. Similar effects were found for monopotassium phosphate (MKP) and K_2_Ca_2_Mg(SO_4_)_4_·2(H_2_O) (polyhalite) in the current research. The results of the sprayed salts were similar for CDM suppression and the previously reported suppression of cucumber PM [7]. Thus, the spraying of the various salts may be aimed at both diseases in cucumber. Under commercial conditions, the achieved CDM reduction was ca. 50%. Similarly, spraying with the salts Ca(NO_3_)_2_, K^−^ phosphite, and K_2_SO_4_ decreased cucumber PM on the leaves of young plants [26]. PM and DM of the cucurbits cucumber and cantaloupe, PM of pepper, and early blight and late blight of tomato were reduced by various treatments including sprays of CaCl_2_, K_2_HPO_4_, and K_2_CO_3_ [27]. MKP was previously reported as a PM suppressor on cucumber, melon, rose, peach, nectarine, and grape [19,28,29,30].

The fact that sprays at specific low molar concentrations and above were similarly effective points to induced resistance in the cucumber–CDM pathosystem, as suggested earlier for cucumber PM [7] and sweet basil downy mildew [8]. Combining two or three salts in a spray treatment with no additive effect as compared with the single salt spray and the fact that all tested salts were effective in CDM control also point to the possibility of an induced resistance mechanism. Induced resistance is a possible mode of action where a basic signal is given by any of the salts on the cucumber leaves, and no additional effect is achieved with a more intensive signal that may be inflicted by a higher concentration or combined salts. Similarly, Abdel-Kader et al. [27] also attributed induced resistance activity to sprayed salts, CaCl_2_, K_2_HPO_4_, and K_2_CO_3_, among other chemical and microbial inducers aimed at cucumber, pepper, and tomato foliar diseases. 

Calcium chloride was tested as an alternative induced resistance agent to benzothiadiazole (BTH) and hydrogen peroxide [31]. Induced resistance in pearl millet against DM (*Sclerospora graminicola*) was obtained by BTH, CaCl_2_, and H_2_O_2_ seed treatments. Seedlings that were grown from the treated seeds demonstrated increased hypersensitive response in reaction to inoculation with *S. graminicola* [31]. Additionally, Hamza et al. [32] reported on KH_2_PO_4_, K_2_HPO_4_, and MgSO_4_ as chemical inducers of resistance, among others, sprayed on cucumber plants for the control of PM in a greenhouse. Similarly, Reuveni et al. [19,28,29] demonstrated that phosphates and K salts (MKP) were appropriate for use as inducing foliar fertilizers that suppress cucumber PM. The more studied element for its induced resistance effect is Si. For instance, in cucumber, the effects of Si on the activities of defense-related enzymes were studied in the presence of CDM, revealing an increase in the activities of several defense-related enzymes [33] and pointing to the same conclusion of possible induced resistance in cucumber DM and PM.

## 4. Materials and Methods

### 4.1. General

Some of the sites, plants, and treatment methods were described recently [7]. Cucumber (*Cucumis sativus*) seeds were germinated planted in seedlings trays containing cells of 4 × 4 × 10 cm inverted pyramidal cells filled with perlite (medium size, 1.2 mm, Agrifusia, Fertilizers & Chemicals Ltd., Haifa, Israel) for pot experiments. Developed seedlings were transplanted at the age of 3 weeks to pots as described below; otherwise, seeds were germinated directly in the pots described below. To prevent damping off in the pots harboring germinating seeds, we treated them once with a fungicide (0.25% Dynon in water drench, containing 722 g/L Propamocarb HCl, Bayer, Germany). Cucumber cv. Bet Alpha, susceptible to CDM, was used in pot experiment sites A and B as described below. Cv. 501 susceptible to CDM but is partially resistant to cucumber powdery mildew (*Posdosphaera xanthii*) was used in the commercial-like experiments described below for site C.

The experiments involving potted plants were performed at two sites in Israel: the Volcani Institute in Rishon LeZion (Site A) and the Gilat Research Center of the Volcani Institute in the northern Negev (Site B). Experiments were also carried out using plants grown in containers under commercial greenhouse conditions at a net house, Gilat Research Center (Site C). At Site A, experiments were carried out in 2 L pots. At Site B, experiments were carried out in 2 L pots. The commercial-like experiments (Site C) were carried out in 136 L containers. For the fertigation experiments, cucumber plants were planted in pots or containers filled with perlite. For the foliar salt treatments experiments, we used a potting mixture consisting of coconut fiber and tuff (unsorted to 8 mm; 7:3 *v*/*v*). Depending on the season, the plants were irrigated to excess via a drip system two to four times a day, at a volume calibrated to lead to >30% water leaching. After analyzing the irrigation and drainage solutions once every 2 weeks, the daily irrigation volume was determined to prevent over-salinization or acidification of the root-zone solution. Plants in pots and containers were maintained according to the local extension service’s recommendations. All pot experiments were irrigated with freshwater (electrical conductivity (EC) = 2.2 dS/m). As described below, the tested elements were applied with the irrigation water (fertigation) or as a foliar spray, summarized in Table 1 and reported earlier [7].

### 4.2. Effects of Different Concentrations of N, P, K, Ca, and Mg in the Fertigation Solution on CDM (Expts B-#-f)

Pot experiments were conducted in an unheated, polyethylene-covered greenhouse at Site B. These experiments aimed to study the effects of different N, P, K, Ca, and Mg concentrations in the fertigation solution (“f” treatments) on the development of CDM in potted cucumber plants. The cucumber plants were planted in 2 L perlite-filled pots with one plant per pot, in 10 replicates, and each set of cation concentrations was repeated twice. Plants did not bear fruits.

Nutrient solutions were prepared in 500 L containers containing all of the added nutrients. All of the plants were fertigated with 5–3–8 (N–P_2_O_5_–K_2_O) fertilizer (Fertilizers and Chemical Compounds Ltd., Haifa, Israel) for 2 weeks until the establishment of plants. Later on, the effect of cation concentration was tested by tailoring the fertigation solution to each N, P, K, Ca, and Mg concentration of interest, as described below and earlier [7,8,9,10,11,12]. The concentrations of nutrients that were not part of the experiments and remained the same across all treatments were as follows: 5.7 mM N (90% NO_3_^−^-N and 10% NH_4_^+^-N; excluding Experiment B1-N below), 0.35 mM P (excluding Experiment B1-P below), 2.6 mM K (excluding Experiment B1-K below), 1.3 mM Ca (excluding Experiment B1-Ca below), 0.54 mM Mg (excluding Experiment B1-Mg below), 1.1 mM SO_4_, 0.023 mM B, 9.8 µM Fe, 4.9 µM Mn, 2.1 µM Zn, 0.31 µM Cu, and 0.16 µM Mb. Solutions were prepared by dissolving KH_2_PO_4_, K_2_SO_4_, KNO_3_, NH_4_H_2_PO_4_, NaNO_3_, and NH_4_NO_3_ in water [34]. In Experiment B1-Ca-f, Ca was applied as CaCl_2_, and, in Experiment B1-Mg-f, Mg was applied as MgCl_2_. 

N concentration in the fertigation solution (Experiments B-N-f): These experiments aimed to study the effect of the N concentration in the fertigation solution on the development of CDM in potted cucumber plants. To characterize the response of cucumber plants to different concentrations of N in the fertigation solutions, five N concentrations (0.7, 1.4, 2.9, 5.0, 7.1, and 14.3 mM) were used, while the concentrations of the other nutritional elements were kept constant. The EC in the different N-f treatments was 1.03–1.36 dS/m, and the pH of the fertigation solution was 6.99–7.42. 

P concentration in the fertigation solution (Experiments B-P-f): These experiments aimed to study the effect of the P concentration in the fertigation solution on CDM development in potted cucumber plants. To characterize the response of cucumber plants to different concentrations of P in the fertigation solutions, three P concentrations (0, 0.065, and 0.645 mM) were used, while the concentrations of the other nutritional elements were kept constant. The EC in the different P-f treatments was 1.03 and 1.36 dS/m, and the pH of the fertigation solution was 6.99 and 7.32.

K concentration in the fertigation solution (Experiments B-K-f): These experiments aimed to study the effect of the K concentration in the fertigation solution on the development of CDM in potted cucumber plants. To characterize the response of cucumber plants to different concentrations of K in the fertigation solutions, four K concentrations (0.5, 1.0, 1.8, and 5.1 mM) were used, while the concentrations of the other nutritional elements were kept constant. The EC in the different K-f treatments was 1.06–1.36 dS/m, and the pH of the fertigation solution was 7.05–7.62.

Ca concentrations in the fertigation solution (Experiment B-Ca-f, CaCl_2_ supplement): These experiments aimed to study the effect of Ca concentration on the development of CDM in potted cucumber plants. To characterize the response of cucumber plants to different concentrations of Ca in the fertigation solutions, four Ca concentrations (1.0, 2.0, 3.0, and 4.0 mM) were used, while the concentrations of the other nutritional were kept constant. The EC in the different B-Ca-f treatments was 0.99–2.50 dS/m, and the pH of the fertigation solution was 6.43–7.12.

Mg concentrations in the fertigation solution (Experiment B-Mg-f, MgCl_2_ supplement): These experiments aimed to study the effect of the Mg concentration in the fertigation solution on the development of CDM in potted cucumber plants. To characterize the response of cucumber plants to different concentrations of Mg, four Mg concentrations (0, 1.65, 3.29, and 4.94 mM) were used, while the concentrations of the other nutritional elements were kept the same for all treatments. The EC in the different B-Mg-f treatments was 0.84–1.78 dS/m, and the pH of the fertigation solution was 5.62–6.95.

### 4.3. Foliar Application of Salt Solutions to Potted Cucumber Plants (Expts A-s 1 and B-s 1 to 3)

Cucumber seedlings were transplanted into pots containing growth mixture or perlite, as described above, unless noted otherwise. Throughout the experiments, fertigation was carried out with the fertilizer 4–2–6 (N–P_2_O_5_–K_2_O) + 3% microelements (Fertilizers and Chemical Compounds Ltd., Haifa, Israel). In young cucumber plants, following the formation of two to three leaves, when plants in the experiments reached 20 to 30 cm in height (Expts B-s), and in mature plants bearing at least 18 leaves (Expts A), the foliar treatment was initiated, and plants were artificially inoculated with sporangia of *Pseudoperonospora cubensis*. Experiments A-s 1 and exp. B-s 1 to 3 were repeated twice with five replicates each.

Sprays were conducted twice a week with a hand sprayer forming a small-drop mist and containing up to 1 L water. Solutions contained 0.1% and 1.0% of K_2_SO_4_, KCl, MgCl_2_, and CaCl_2_ reaching various molar concentrations of the applied ions (Exp. A-s1), NaCl at concentrations of 0.025, 0.05, 0.1, 0.2, 0.4, and 0.6 M (Cl^−^) (Exp. B-s1), NaCl at concentrations of 0.025, 0.05, 0.1, 0.2, 0.4, and 0.6 M (Cl^−^ or Na^+^) (Exp. B-s2), and K_2_SO_4_, MgSO_4_, KCl, MgCl_2_, and NaCl at anion concentrations of 0.2 M (Expt. B-s3).

### 4.4. Foliar Application of Salt Solutions to Cucumber Plants Grown under Semi-Commercial Conditions (Expts A-SCs a and b)

Cucumber seedlings were transplanted into 10 L pots containing growth mixture and located in an experimental greenhouse in site A. Fertigation was carried out with the fertilizer 4–2–6 (N–P_2_O_5_–K_2_O) + 3% microelements (Fertilizers and Chemical Compounds Ltd., Haifa, Israel) throughout the experiments. The foliar treatments were initiated when plants were artificially inoculated with the inoculum of *P. cubensis* over mature plants bearing at least 20 leaves, and flowers and fruits. Experiments were conducted twice with six replicates and arranged in randomized blocks.

Spray treatments were applied twice a week with a hand sprayer containing up to 1 L of water and forming a mist of tiny droplets. The experiments included sprays with the salts MgCl_2_, CaCl_2_, and K_2_SO_4_ at concentrations of 0.05 and 0.1 M (anion) and a combination of MgCl_2_ 0.05 M (Cl), CaCl_2_ 0.05 M, and K_2_SO_4_ 0.1 M (SO_4_), resulting in a total of 0.1 M Cl and SO_4_ (Expt. A-CSs a). The treatments of monopotassium phosphate (MKP, Fertilizers and Chemical Compounds Ltd., Haifa, Israel) were conducted at 1% as recommended, while those of polyhalite (K_2_Ca_2_Mg(SO_4_)_4_·2(H_2_O) (polysulfate, Fertilizers and Chemical Compounds Ltd., Haifa, Israel) [35] were conducted at concentrations of 0.125% and 0.25% (Expt. A-SCs b). A spray with water with no salt served as an untreated control.

### 4.5. Commercial-Like Net House Experiments (CL) to Test Fertigation through Irrigation and Spray

At Site C, experiments were carried out in a net-covered greenhouse (net = 50 mash). One experiment was performed to test a spray treatment of two salts combined (Expt. CL1), and the second experiment tested five spray treatments (Expt. CL2). Cucumber plants (cv. 501, Hazera Genesis, Israel) were maintained in a commercial nursery (Shetil Neto, Gevaram, Israel), transplanted on 1 March 2019 to the perlite boxes mentioned below and grown as fruit-bearing plants until the end of May 2019.

Cucumber plants were planted in perlite (medium size, 1.2 mm, Agrifusia) growth medium in polystyrene containers (1.0 × 0.8 × 0.17 m), six plants per container. Plants were irrigated daily according to local extension service recommendations. During the initial 5 days, plants were sprinkler-fertigated with 4.3 mM N (10% NH_4_), 1.6 mM K, and 0.65 mM P in the fertigation solution to aid their establishment. Experiment CL1 consisted of the spray treatments: water and salt combination of MgCl_2_ + K_2_SO_4_ 0.1 M (Cl) + 0.1 M (SO_4_). Experiment CL2 consisted of the sprays of MgCl_2_ 0.1 M (Cl^−^), K_2_SO_4_ 0.1 M (SO_4_), a combination of MgCl_2_ + K_2_SO_4_, and 1% MKP. In both experiments, spray treatments were carried out with a backpack sprayer equipped with a conical nozzle. Sprays were administrated until runoff, every 3 to 6 days. Experiment CL1 was organized as randomized split plots with four replicates for each combination of treatments, and experiment CL2 was organized in randomized blocks with four replicates. Each plot consisted of two perlite boxes containing six plants in one box, in two rows; thus, each plot consisted of 12 plants.

### 4.6. Pathogen and Disease

Cucurbit downy mildew (*Pseudoperonospora cubensis*, CDM) occurs naturally, and the cucumber cultivars used in the pot experiments and the commercial-like experiments are susceptible to CDM. Nevertheless, to ensure even pathogen distribution across the plants, every pot experiment was artificially infested with sporangia suspension (10^4^ cells/mL) using a hand sprayer admitting a fine mist that dried over within 10 min. The pathogen formed typical symptoms of CDM on leaves, and it was evaluated on a 0–100% severity scale, where 0& = healthy leaves and 100% = leaves completely covered by disease symptoms. Leaves evaluated were numbers 3–5 from the plant base as mature leaves, numbers 6–9 as medium leaves, and numbers 9–12 as younger leaves. For whole-plant CDM severity, an average of evaluated leaves from each plant replicate was calculated. Values of the area under the disease progress curve (AUDPC) were calculated throughout epidemic development.

### 4.7. Element Analysis

Fully expended mature leaves were sampled randomly in all experiments at harvest time from potted plants and the semi-commercial (large pots) and the commercial-like (containers) experiments and used to determine mineral concentration. The shoots were rinsed with distilled water and dried in an oven at 70 °C for 48 h. The dried plant material was ground and subjected to chemical analysis. N, P, and K concentrations of the shoots were analyzed after digestion with sulfuric acid and peroxide [36]. Ca, Na, and Mg concentrations were analyzed after digestion with nitric acid and perchlorate [37]. The concentrations of N and P were determined with an autoanalyzer (Lachat Instruments, Milwaukee, WI, USA). K, Na, Mg, and Ca were analyzed with an atomic absorption spectrophotometer (Atomic absorption Perkin-Elmer 460, Norwalk, CT, USA). The Cl was extracted from the shoot in water (100:1 water and dry matter) and determined with a chloride analyzer (model 926, Sherwood Scientific, Cambridge, UK). 

### 4.8. Statistics

Data in percentages were arcsine-transformed before further analysis. The area under the disease progress curve (AUDPC) values were calculated. Standard errors of the mean (SEs) were calculated and presented alongside the number of degrees of freedom (DF = n − 1 for controlled conditions experiments and DF = n − 2 for correlations calculated for field conditions). The disease severity and AUDPC data were analyzed by ANOVA and Tukey-Kramer HSD test. Statistical analysis was performed (α = 5%) using JMP 14.0 software (SAS Institute, Cary, NC, USA). 

To test the relationship between the concentration of a nutritional element and values of disease severity in experimental replicates (plots), correlation coefficients were calculated, and the best line formula was calculated using all individual pairs of data (n). Correlation line types included linear, exponential, logarithmic, and polynomial. The regression formulas and the Pearson correlation coefficient values (*r*) were presented along with significance levels (*p*) according to the degrees of freedom (n − 2) in the relevant figures captions.

## 5. Conclusions

Mg and P in the irrigation water of cucumber plants and the spray of various salts decreased CDM severity. Salts of different types may be used as part of integrated disease management in cucumber plants. It is suggested that the cucumber plants react via induced immunity to the ions reaching their surfaces, as also described for cucumber PM [7]. Further induced resistance research is currently being pursued with tomato plants (Maya Bar, Yigal Elad, Uri Yermiyahu, and coworkers, The Volcani Institute, Rishon LeZion, Israel).

## Figures and Tables

**Figure 1 plants-11-01007-f001:**
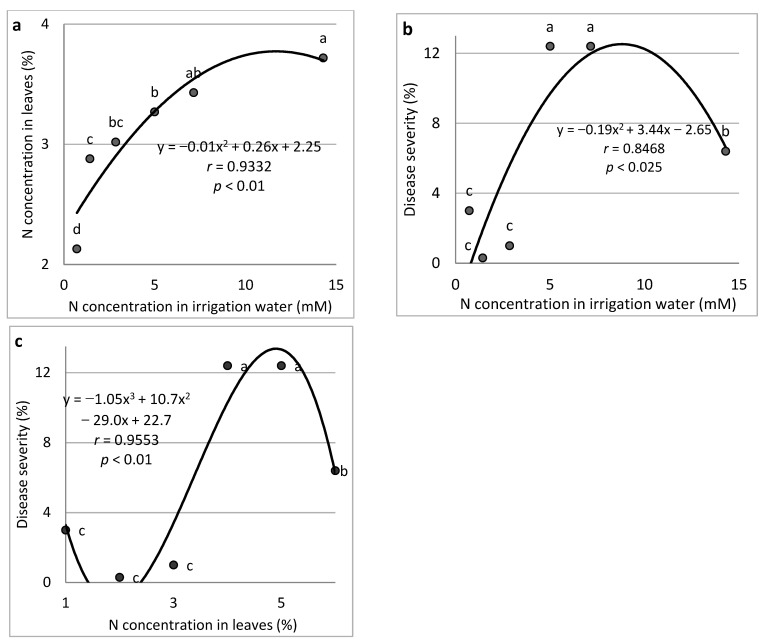
Effect of N concentration treatments in the irrigation water on cucumber downy mildew (CDM, *Pseudoperonospora cubensis*). N concentration in the leaves is presented in relation to N concentration in the irrigation water (**a**), CDM severity is presented in relation to N concentration in the irrigation water (**b**), and CDM severity is presented in relation to the N concentration in the leaves (**c**). The disease was evaluated on a 0 to 100% severity scale, where 0% = healthy leaves and 100% = leaves completely covered by CDM symptoms. Values of the area under the disease progress curve (AUDPC) were calculated for 18 days after disease onset. In each graph, values for each N treatment followed by a common letter are significantly not different from each other according to Tukey-Kramer’s HSD test (*p* ≤ 0.05). The regression formulas of the drawn trend lines are presented, and the correlation Pearson regression values (*r*) are presented along with significance levels (*p*).

**Figure 2 plants-11-01007-f002:**
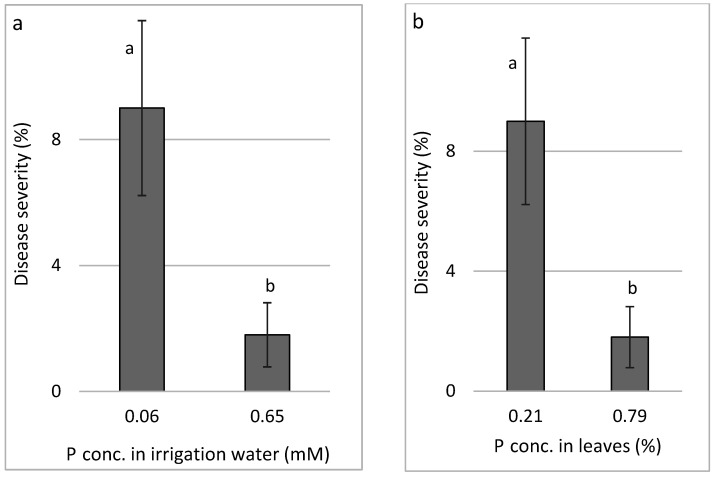
Effect of P concentration treatments in the irrigation water on cucumber downy mildew (CDM, *Pseudoperonospora cubensis*). CDM severity is presented in relation to P concentration in the irrigation water (**a**) and in relation to the P concentration in the leaves (**b**). The disease was evaluated on a 0 to 100% severity scale, where 0% = healthy leaves and 100% = leaves completely covered by CDM symptoms. In each graph, values for each P treatment followed by a common letter are significantly not different from each other according to Tukey-Kramer’s HSD test (*p* ≤ 0.05).

**Figure 3 plants-11-01007-f003:**
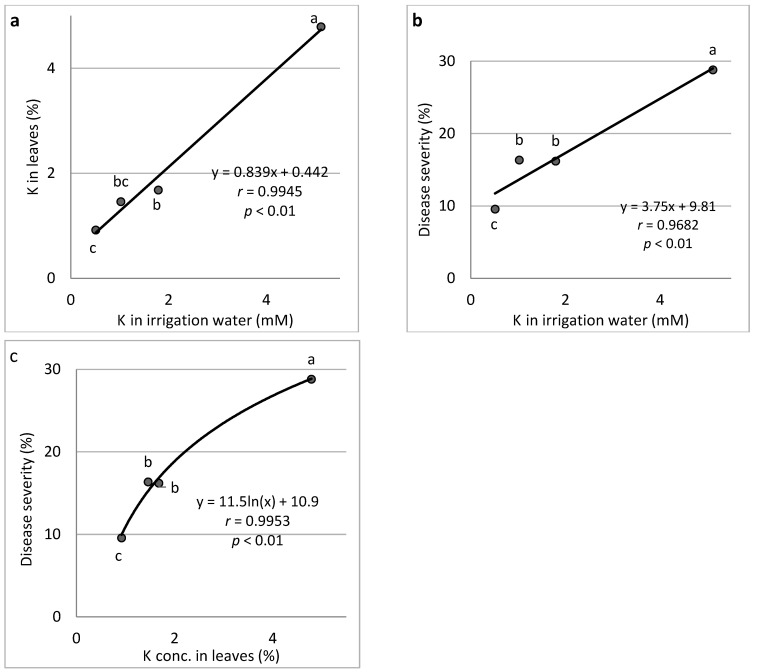
Effect of K concentration treatments in the irrigation water on cucumber downy mildew (CDM, *Pseudoperonospora cubensis*). K concentration in the leaves is presented in relation to K concentration in the irrigation water (**a**), CDM severity is presented in relation to K concentration in the irrigation water (**b**), and CDM severity is presented in relation to the K concentration in the leaves (**c**). The disease was evaluated on a 0 to 100% severity scale, where 0% = healthy leaves and 100% = leaves completely covered by CDM symptoms. Values of the area under the disease progress curve (AUDPC) were calculated for 18 days after disease onset. In each graph, values for each K treatment followed by a common letter are significantly not different from each other according to Tukey-Kramer’s HSD test (*p* ≤ 0.05). The regression formulas of the drawn trend lines are presented, and the correlation Pearson regression values (*r*) are presented along with significance levels (*p*).

**Figure 4 plants-11-01007-f004:**
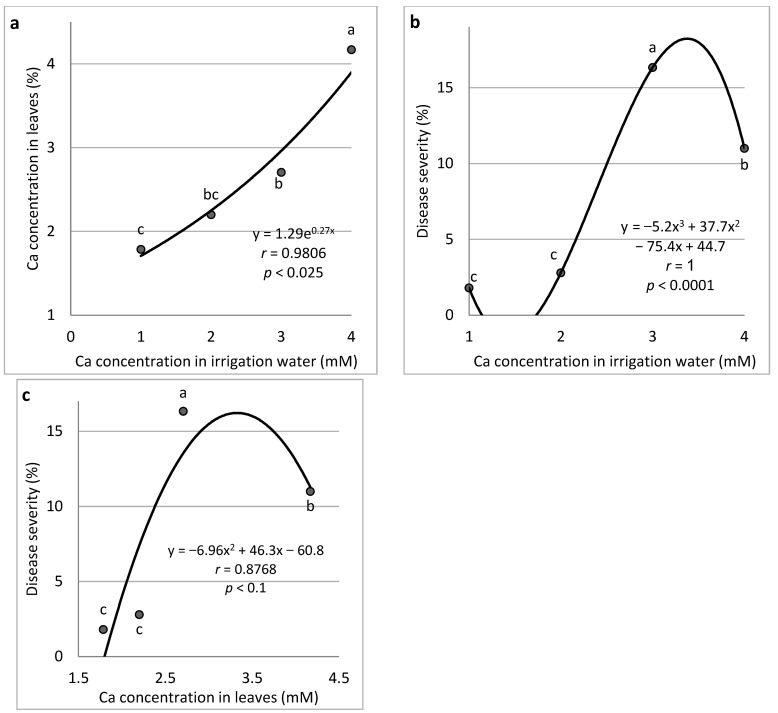
Effect of Ca concentration treatments in the irrigation water on cucumber downy mildew (CDM, *Pseudoperonospora cubensis*). Ca concentration in the leaves is presented in relation to Ca concentration in the irrigation water (**a**), CDM severity is presented in relation to Ca concentration in the irrigation water (**b**), and CDM severity is presented in relation to the Ca concentration in the leaves (**c**). The disease was evaluated on a 0 to 100% severity scale, where 0% = healthy leaves and 100% = leaves completely covered by CDM symptoms. Values of the area under the disease progress curve (AUDPC) were calculated for 18 days after disease onset. In each graph, values for each Ca treatment followed by a common letter are significantly not different from each other according to Tukey-Kramer’s HSD test (*p* ≤ 0.05). The regression formulas of the drawn trend lines are presented, and the correlation Pearson regression values (*r*) are presented along with significance levels (*p*).

**Figure 5 plants-11-01007-f005:**
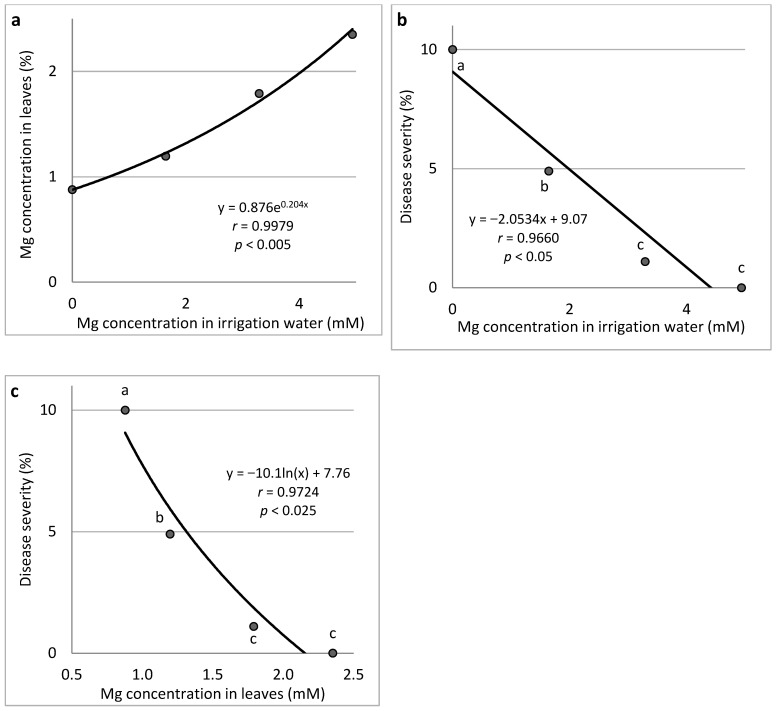
Effect of Mg concentration treatments in the irrigation water on cucumber downy mildew (CDM, *Pseudoperonospora cubensis*). Mg concentration in the leaves is presented in relation to Mg concentration in the irrigation water (**a**), CDM severity is presented in relation to Mg concentration in the irrigation water (**b**), and CDM severity is presented in relation to the Mg concentration in the leaves (**c**). The disease was evaluated on a 0 to 100% severity scale, where 0% = healthy leaves and 100% = leaves completely covered by CDM symptoms. Values of the area under the disease progress curve (AUDPC) were calculated for 18 days after disease onset. In each graph, values for each Mg treatment followed by a common letter are significantly not different from each other according to Tukey-Kramer’s HSD test (*p* ≤ 0.05). The regression formulas of the drawn trend lines are presented, and the correlation Pearson regression values (*r*) are presented along with significance levels (*p*).

**Figure 6 plants-11-01007-f006:**
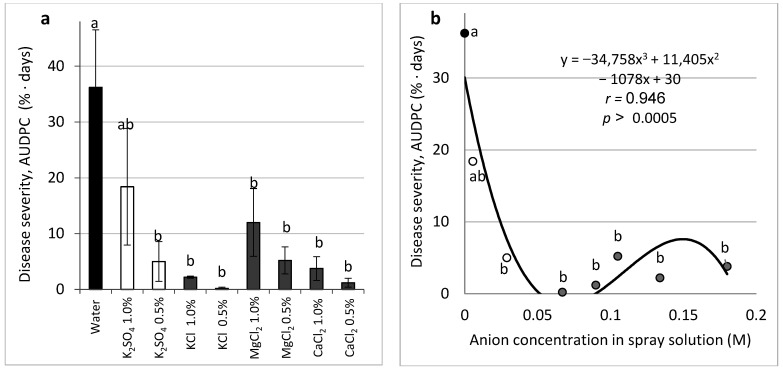
Effect of salts sprays on the severity of cucumber downy mildew (CDM, *Pseudoperonospora cubensis*) on cucumber leaves. Sulfate (empty columns and dots) and chloride salts in two equal percentage weight concentrations were applied as sprays on mature plants twice a week. Disease severity is described per salt treatment (**a**) and according to anion concentration (**b**). CDM severity was evaluated on a 0 to 100% scale, where 0% = no disease symptoms and 100% = leaf fully covered by CDM symptoms. Values of the area under the disease progress curve (AUDPC) were calculated for 18 days since disease onset. Values followed by a common letter are significantly not different from each other according to Tukey-Kramer’s HSD test (*p* ≤ 0.05). Bars = SE. The regression formula of the drawn trend line is presented, and the correlation Pearson regression value (*r*) is presented along with the significance level (*p*).

**Figure 7 plants-11-01007-f007:**
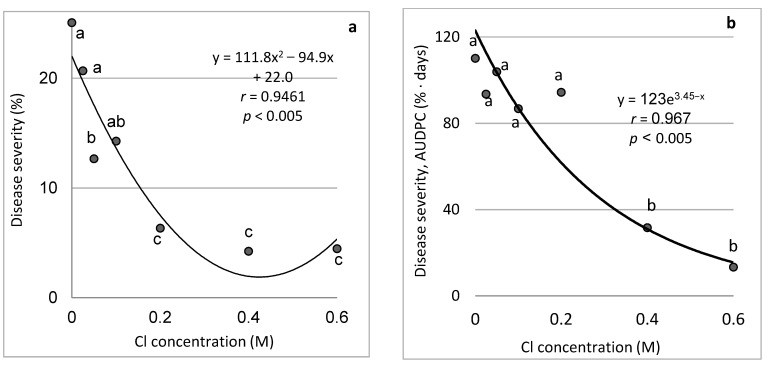
Effect of chloride ion concentration in the spraying solution on the severity of cucumber downy mildew (CDM, *Pseudoperonospora cubensis*) on leaves of cucumber plants. NaCl was applied as spray twice a week. CDM severity was evaluated on a 0 to 100% scale, where 0% = no disease symptoms and 100% = leaf fully covered by CDM symptoms (**a**). The area under the disease progress curve (AUDPC) was calculated (**b**). In each graph, values followed by a common letter are significantly not different from each other according to Tukey-Kramer’s HSD test (*p* ≤ 0.05). The regression formulas of the drawn trend lines are presented, and the correlation Pearson regression values (*r*) are presented along with significance levels (*p*).

**Figure 8 plants-11-01007-f008:**
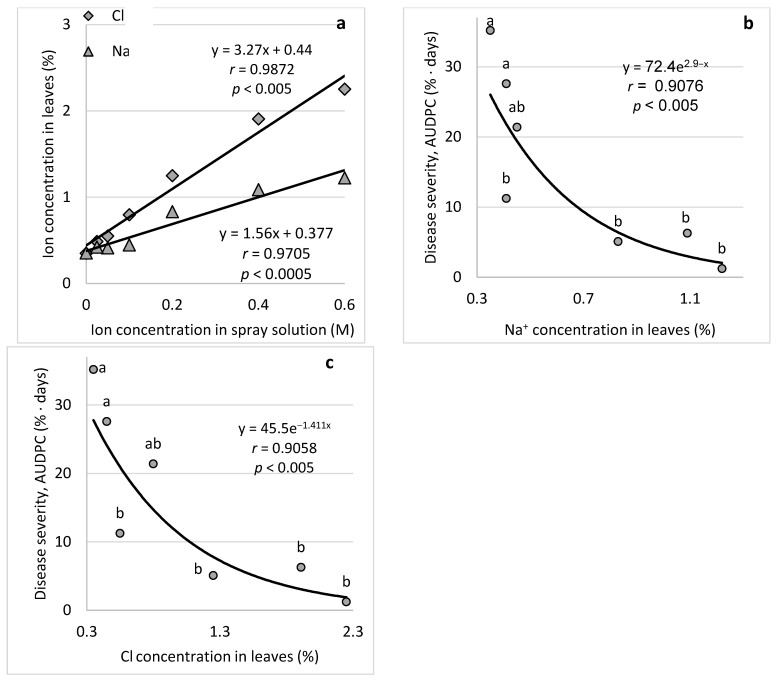
Relations of Na and Cl concentrations in leaves and cucumber downy mildew (CDM, *Pseudoperonospora cubensis*) following spraying of NaCl solutions. Relationship between NaCl concentration in the irrigation water and the ions concentration in leaves (**a**), and effect of leaf Na^+^ concentration (**b**) and Cl^−^ concentration (**c**) on CDM severity. CDM severity was evaluated on a 0–100% scale, where 0% = no disease symptoms and 100% = leaf fully covered by CDM symptoms. The area under the disease progress curve (AUDPC) was calculated. Values followed by a common letter are significantly not different from each other according to Tukey-Kramer’s HSD test (*p* ≤ 0.05). The regression formulas of the drawn trend lines are presented, and the correlation Pearson regression values (*r*) are presented along with significance levels (*p*).

**Figure 9 plants-11-01007-f009:**
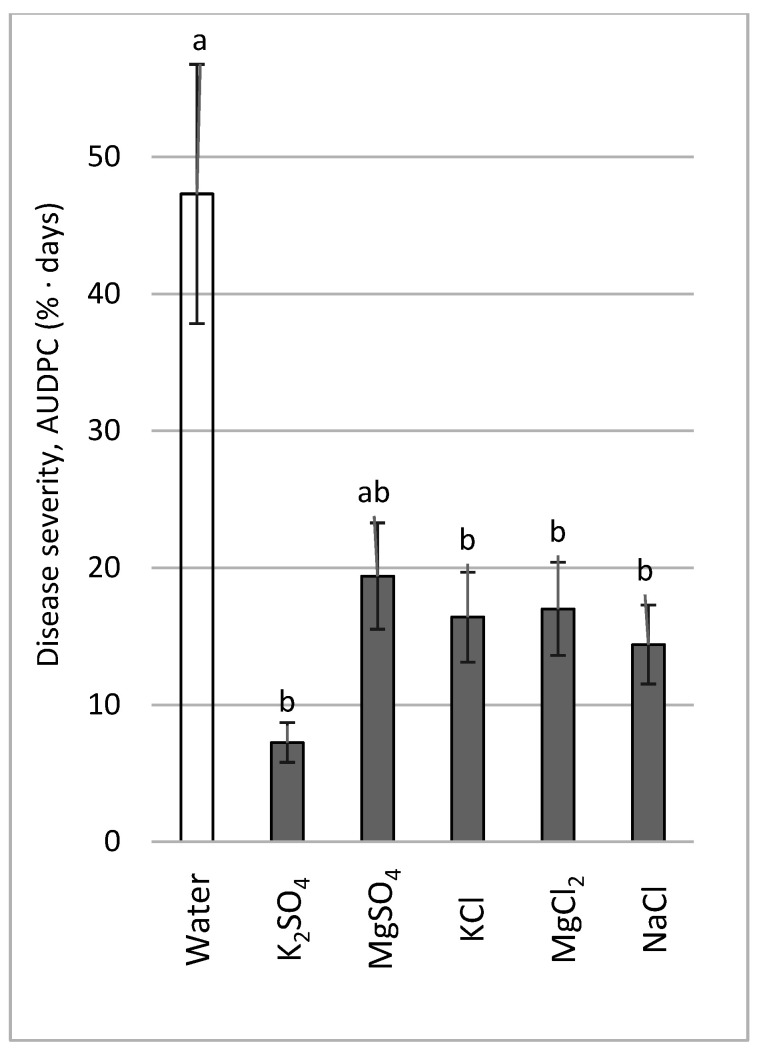
Effect of Cl- and SO_4_-containing salts on cucumber downy mildew (CDM, *Pseudoperonospora cubensis*) after spraying 0.2 M solutions every 14 days, calculated according to the anion. CDM severity was evaluated on a 0 to 100% scale, where 0% = no disease symptoms and 100% = leaf fully covered by CDM symptoms; the area under the disease progress curve (AUDPC) was calculated. Values followed by a common letter are significantly not different from each other according to Tukey-Kramer’s HSD test (*p* ≤ 0.05).

**Figure 10 plants-11-01007-f010:**
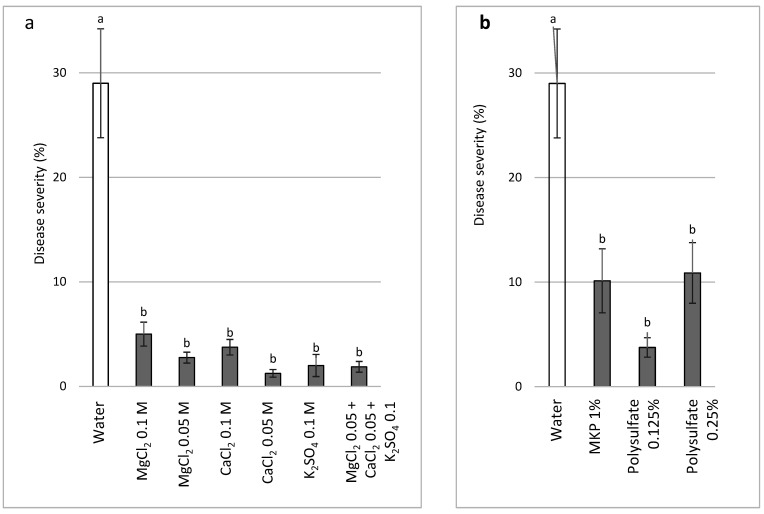
Effect of bi-weekly sprays of (**a**) various salts at two concentrations, individually and in combination (MgCl_2_ 0.05 M (Cl), CaCl_2_ 0.05 M (Cl), and K_2_SO_4_ 0.1 M (SO_4_)) (Expt. A-SCs-b), and (**b**) polyhalite and monopotassium phosphate (Expt. A-SCs-b) on cucumber downy mildew (CDM, *Pseudoperonospora cubensis*) at 61 days after planting. CDM severity was evaluated on a 0 to 100% scale, where 0% = no disease symptoms and 100% = leaf fully covered by symptoms. Values for each date and in each graph followed by a common letter are significantly not different from each other according to one-way ANOVA with Tukey’s HSD (*p* ≤ 0.05). Bars = SE.

**Figure 11 plants-11-01007-f011:**
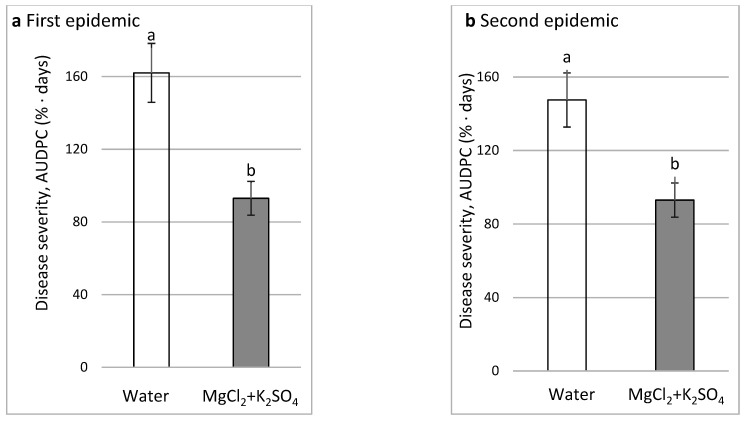
Effect of sprays every 3–6 days with MgCl_2_ 0.1 M (Cl) + K_2_SO_4_ 0.1 M (SO_4_) on cucumber downy mildew (CDM, *Pseudoperonospora cubensis*) under commercial-like conditions (CL1). CDM was followed during the first epidemic on lower leaves (**a**) and the second epidemic on upper leaves (**b**). CDM severity was evaluated on a 0 to 100% scale, where 0% = no disease symptoms and 100% = leaf fully covered by CDM symptoms. The area under disease progress curves (AUDPCs) through 30 (**a**) and 45 (**b**) days were calculated. Values in each graph followed by a common letter are significantly not different from each other according to one-way ANOVA with Tukey’s HSD (*p* ≤ 0.05).

**Figure 12 plants-11-01007-f012:**
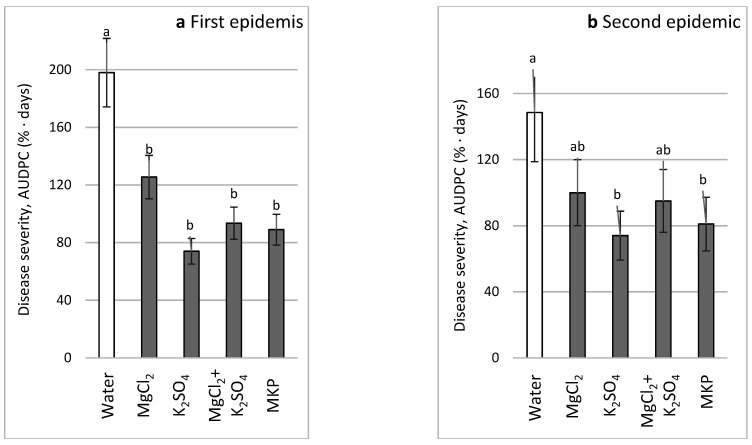
Effect of sprays every 3–6 days with MgCl_2_ 0.1 M (Cl), K_2_SO_4_ 0.1 M (SO_4_), MgCl_2_ + K_2_SO_4_, or monopotassium phosphate (MKP) on cucumber downy mildew (CDM, *Pseudoperonospora cubensis*) under commercial-like conditions (CL2). CDM was followed during the first epidemic on lower leaves (**a**) and the second epidemic on upper leaves (**b**). CDM severity was evaluated on a 0 to 100% scale, where 0% = no disease symptoms and 100% = leaf fully covered by CDM symptoms. The area under disease progress curves (AUDPCs) through 30 (**a**) and 45 (**b**) days were calculated. Values in each graph followed by a common letter are significantly not different from each other according to one-way ANOVA with Tukey’s HSD (*p* ≤ 0.05).

**Table 1 plants-11-01007-t001:** Experimental setup, factors tested, application methods, and growing seasons.

Site	Code	Growing Setting	Elements Tested	Application	Season
B	B-N-f	Pots	N	Fertigation (f)	All year
B	B-P-f	Pots	P	Fertigation	All year
B	B-K-f	Pots	K	Fertigation	All year
B	B-Ca-f	Pots	Ca (Cl)	Fertigation	All year
B	B-Mg-f	Pots	Mg (Cl)	Fertigation	All year
A	A-s1	Pots	K_2_SO_4_, KCl, MgCl_2_, CaCl_2_	Foliar (spray, “s”)	All year
B	B-s1	Pots	NaCl	Foliar	All year
B	B-s2	Pots	NaCl	Foliar	All year
B	B-s3	Pots	MgSO_4_, MgCl_2_, K_2_SO_4_, KCl, NaCl	Foliar	All year
A	A-SCs-a	Large pots (semi-commercial)	MgCl_2_, CaCl_2_, K_2_SO_4_, MgCl_2_ + CaCl_2_ + K_2_SO_4_	Foliar	Autumn–Winter
A	A-SCs-b	Large pots (semi-commercial)	Monopotassium phosphate (MKP), K_2_Ca_2_Mg(SO_4_)_4_·2(H_2_O) (polyhalite as polysulfate)	Foliar	Winter–Spring
C	CL1	Boxes (commercial-like)	MgCl_2_ + K_2_SO_4_	Fertigation and foliar	Spring
C	CL2	Boxes (commercial-like)	MgCl_2_, K_2_SO_4_, MgCl_2_ + K_2_SO_4_, MKP	Foliar	Spring

## Data Availability

The data that support the findings of this study are available from the corresponding author upon reasonable request.

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
