# Peer review of "Effect of Mineral Nutrition and Salt Spray on Cucumber Downy Mildew (Pseudoperonospora cubensis)"

_plants, 2022, doi:10.3390/plants11081007_

Round 1

Reviewer 1 Report

 Manuscript title: Effect of mineral nutrition and salts spray on cucumber downy mildew (Pseudoperonospora cubensis)
Manuscript ID: plants-1643928
Journal: Plants
General comments:
The main objectives of the present manuscript were to examined the effects of N, P, K, Ca and Mg, applied in the  irrigation water and the effect of sprays with salts containing the cations Ca, Mg, and K 69 and the anions Cl and SO4 on CDM. 

Abstract is informative and gave me direct idea of the obtained results. In introduction section, the authors are requested to update the old used references especially there are new published articles in the field of study. Material and methods contain details which help other researchers to follow. After reading the discussion section, I suggest the authors to not repeat some sentences were already mentioned in the introduction section.

Author Response

Thank you for the review. The corrections were made and are in the file.

Reviewer 2 Report

The manuscript (MS) deals with the effect of mineral nutrition and salt spray on cucumber downy mildew (Pseudoperonospora cubensis). This topic is of high interest due to the potential of mineral nutrition and salt spray as part of the integrated plant protection management to maintain plant health and reduce yield losses.

In general, the MS is written very well with minor problems in English language and scientific soundness (see MS). The authors should add the charge (+ an -) to the ions in the text, some of them have it, some not. Furthermore, the graphs layout should be standardised, for instance, use the same letter size, figure letter at the same spot outside the graph (maybe with bracket to distinguish them from the significant letters), figure legend outside the graph, try to position the formula and the other data in same corner in each graph, try to lower the numbers in the chemical compounds.

More notes and recommendations can be found in the manuscript.

Nevertheless, I am confident that after minor revision by the authors the MS will be in good shape for publication in Plants. Therefore, I recommend ‘minor revision’ of the reviewed MS.

Author Response

Reviewer 2

Thank you for the remarks and file correction. The ms was corrected accordingly. The details of the response are in the attached file.
